# Fuzzy K-Nearest Neighbor Based Dental Fluorosis Classification Using Multi-Prototype Unsupervised Possibilistic Fuzzy Clustering via Cuckoo Search Algorithm

**DOI:** 10.3390/ijerph20043394

**Published:** 2023-02-15

**Authors:** Ritipong Wongkhuenkaew, Sansanee Auephanwiriyakul, Nipon Theera-Umpon, Kasemsit Teeyapan, Uklid Yeesarapat

**Affiliations:** 1Department of Computer Engineering, Faculty of Engineering, Chiang Mai University, Chiang Mai 50200, Thailand; 2Graduate School, Chiang Mai University, Chiang Mai 50200, Thailand; 3Department of Computer Engineering, Faculty of Engineering, Excellence Center in Infrastructure Technology and Transportation Engineering, Biomedical Engineering Institute, Chiang Mai University, Chiang Mai 50200, Thailand; 4Department of Electrical Engineering, Faculty of Engineering, Biomedical Engineering Institute, Chiang Mai University, Chiang Mai 50200, Thailand; 5Biomedical Engineering Institute, Chiang Mai University, Chiang Mai 50200, Thailand; 6Empress Dental Care Clinic, 224/1 M.12 Klong-Chonlaprathan Road, Nongkwai, Hangdong, Chiang Mai 50230, Thailand

**Keywords:** dental fluorosis, Dean’s index, possibilistic, c-means clustering, cuckoo search, Lévy flights

## Abstract

Dental fluorosis in children is a prevalent disease in many regions of the world. One of its root causes is excessive exposure to high concentrations of fluoride in contaminated drinking water during tooth formation. Typically, the disease causes undesirable chalky white or even dark brown stains on the tooth enamel. To help dentists screen the severity of fluorosis, this paper proposes an automatic image-based dental fluorosis segmentation and classification system. Six features from red, green, and blue (RGB) and hue, saturation, and intensity (HIS) color spaces are clustered using unsupervised possibilistic fuzzy clustering (UPFC) into five categories: white, yellow, opaque, brown, and background. The fuzzy k-nearest neighbor method is used for feature classification, and the number of clusters is optimized using the cuckoo search algorithm. The resulting multi-prototypes are further utilized to create a binary mask of teeth and used to segment the tooth region into three groups: white–yellow, opaque, and brown pixels. Finally, a fluorosis classification rule is created based on the proportions of opaque and brown pixels to classify fluorosis into four classes: Normal, Stage 1, Stage 2, and Stage 3. The experimental results on 128 blind test images showed that the average pixel accuracy of the segmented binary tooth mask was 92.24% over the four fluorosis classes, and the average pixel accuracy of segmented teeth into white–yellow, opaque, and brown pixels was 79.46%. The proposed method correctly classified four classes of fluorosis in 86 images from a total of 128 blind test images. When compared with a previous work, this result also indicates 10 out of 15 correct classifications on the blind test images, which is equivalent to a 13.33% improvement over the previous work.

## 1. Introduction

Dental fluorosis—a condition where the appearance of the tooth enamel changes—results from an excessive intake of fluoride during the tooth development period. Fluorosis occurs in many regions of the world [1,2,3], with a global prevalence of over 20 million cases as estimated by [4]. Although fluoride is known for its benefits in preventing dental caries, its use requires a balance between caries protection and the risk of dental fluorosis. It is evident from the existing literature that excessive fluoride exposure can take place due to the consumption of fluoride-contaminated groundwater [1]. Occurrences of fluorosis also depend on various factors, such as the dose, the duration of exposure, individual health conditions, and so on. The swallowing of fluoridated toothpastes is also a risk factor for fluorosis in young children under the age of six years [5]. Many studies have found that the disease is most prevalent between 5 and 8 years of age, with both genders equally affected [6].

Although mild dental fluorosis does not degrade the health of teeth, the aesthetic appearance of teeth is commonly a major concern in many people. Young people may make negative psychosocial judgements of other young people based on their enamel appearance [7]. It has been reported that dental fluorosis can diminish their happiness and self-confidence. In addition, several negative attributes—such as being seen as less attractive, less clean, less healthy, less intelligent, less reliable, and less social—are attached to people with severe fluorosis compared to normal people [8,9]. Moreover, children with severe fluorosis can also experience significant psychosocial suffering [10]. Fortunately, treatments for dental fluorosis exist—such as microabrasion in mild cases, or tooth restoration in severe cases (tooth grinding with composite filling, composite veneer, or ceramic veneer)—and can significantly improve patients’ quality of life [11]. Traditionally, the diagnosis of dental fluorosis relies upon a visual examination of teeth, together with various pathological grading systems for fluorosis. The most commonly used classification system is Dean’s index [12], developed by H. T. Dean in 1934, which divides the cosmetic deviations of the teeth into six levels, as described in Table 1. Other alternative indices, such as the Thylstrup-–Fejerskov (TF) index [13], are also widely used.

In the dental fluorosis screening process, clinical grading of tooth enamel is subject to subjective biases by examiners due to various factors—for example, knowing the vicinity where a subject resides possibly hints at the subject’s fluoridation status. Such a bias can be addressed by using a standardized image-based method, where examiners remotely evaluate clinical photographs without prior knowledge of the subject’s location. However, the examination of clinical photographs is still prone to biases, because individual examiners inherently choose different grading thresholds. As a result, there has been interest in developing an image-based automated system for examining the severity of fluorosis. One of the pioneering works on dental fluorosis image analysis is by Pretty et al. [15], where the authors utilized image processing techniques to quantify dental fluorosis levels in fluorescence imaging and experimentally showed that the quantity has a good correlation with the TF index. McGrady et al. [16] also further showed that populations with different levels of fluoride exposure could be discriminated by fluorescence imaging. A dual-camera system that can simultaneously capture both a fluorescence image and a polarized white-light image was used by Liu et al. [17] as a part of an automatic fluorosis classification system. In their work, both image modalities were used to extract five-dimensional fluorosis feature vectors, and then TF index predictions were obtained by RUSBoost [18], with a decision tree as a base learner.

The existing literature on automatic fluorosis classification typically relies on fluorescence images acquired from a quantitative light-induced fluorescence (QLF) imaging device, as it can measure the percentage of fluorescence change in demineralized enamel, which becomes lower due to fluorosis. Other approaches based on techniques such as Raman spectroscopy [19] can also be found in the literature. However, those pieces of specialty equipment come with a disadvantage, due to their non-ubiquity. Alternatively, handheld digital cameras—including smartphone cameras—are more accessible to the public. According to [20], photographic assessments of dental fluorosis, where the photos were taken using a digital SLR camera, exhibited good agreement with clinical assessments using Dean’s index. As a result, it is intuitively tempting to develop an automated dental fluorosis classification tool to assess the severity of dental fluorosis in photographic images of teeth. Such a device could help reduce examiners’ workload in the screening process, which is normally time-consuming, laborious, and prone to human error.

Although photographic image analysis has been introduced to solve some dentistry problems such as dental plaque detection [20,21,22,23,24,25,26] in the past, a few studies have been conducted on the automatic dental fluorosis classification of photographic images. One example includes the work by Yeesarapat et al. [14], where the authors proposed an image-based dental fluorosis classification system using multi-prototype fuzzy c-means (FCM) to classify dental fluorosis into four classes adapted from Dean’s index: Normal, Fluorosis Stage 1, Fluorosis Stage 2, and Fluorosis Stage 3. Their Normal, Fluorosis Stage 2, and Fluorosis Stage 3 classes correspond to Normal, Moderate, and Severe in Dean’s index, respectively, while Fluorosis Stage 1 was created to cover three classes: Questionable, Very Mild, and Mild, as summarized in Table 1. FCM was used to cluster six-dimensional pixel values in the red, green, and blue (RGB) and hue, saturation, and intensity (HIS) color spaces for each of three groups: normal white (either white or yellow), opaque white, and brown pixels. A total of 1600 prototypes were obtained, and the nearest prototype classifier was used to assign each pixel value to one of these three groups. The classification criteria based on the amount of pixel values found in each group were then applied to classify an image of tooth enamel into four dental fluorosis conditions. Their proposed method yielded a correct classification rate of 42.85% for the training set and 53.33% for the blind test set. However, the process required dentists to manually select the tooth regions in the image. Their model selection was also performed manually, and the number of clusters was suboptimal.

Inspired by the work by Yeesarapat et al. [14], this paper proposes an image-based automatic system for dental fluorosis classification based on image segmentation through multi-prototype unsupervised possibilistic fuzzy clustering (UPFC). Feature vectors composed of pixel values in the RGB and HSI color spaces were clustered into five classes: white, yellow, opaque, brown, and background, where the number of clusters was optimized using the cuckoo Search (CS) algorithm. A set of pixels from seven images were used for training, and the best prototypes were chosen by 10-fold cross-validation. After that, tooth segmentation was implemented based on these clusters through the fuzzy k-nearest neighbor (FKNN) method. The proportions of opaque and brown pixels in the tooth region were used to determine four stages of fluorosis: Normal, Stage 1, Stage 2, and Stage 3, with the same severity levels defined by Yeesarapat et al. [14]. We evaluated our proposed method on 128 blind test images taken from 128 subjects in both segmentation and classification tasks. In overall, the proposed system provided a correct dental fluorosis classification for 86 out of 128 images, which is 13.33% better than that obtained in the prior work [14].

Although this image-based fluorosis detection system does not include other tooth deficiency factors—for example, mineralized tissue loss causing cavitation—the dentist can use this system in a pre-screening process to grade the condition before treating the patient. According to the dentist, the treatment rules based on the tooth condition according to the Table 1 are as follows:If there are some opaque white areas (Stage 1 fluorosis), the treatment can be whitening or microabrasion.If there are a lot of opaque white areas and some brown areas (Stage 2 fluorosis), the treatment can be tooth restoration, e.g., tooth grinding with composite filling, composite veneer, or ceramic veneer.If there are a lot of brown areas (Stage 3 fluorosis), the treatment can be similar to the Stage 2 treatment. However, the ceramic used in the treatment might need to be more opaque to cover all of the underlying color. In addition, a restorative dentistry specialist is needed in this case, rather than a general practice dentist.

However, this system can be used in rural areas where there are not enough dentists. Moreover, the pictures can be taken by non-professional personnel with no mouth retractor or the need for a process to dry the teeth. If any mild/severe cases are detected, those patients can be sent to the dentist to recheck for a definitive diagnosis of their teeth before the treatment.

The rest of this paper is organized as follows: Section 2 describes the related backgrounds—HSI color space, UPFC, CS, and FKNN. The dataset and the pipeline of our proposed method (i.e., clustering, segmentation, and classification) are described in Section 3, followed by the experimental results and discussions in Section 4. Finally, our concluding remarks are provided in Section 5.

## 2. Relevant Background

This section describes the four main ingredients of the proposed dental fluorosis classification system, including the transformation of the RGB color space to the HSI color space, the unsupervised possibilistic fuzzy clustering algorithm, the cuckoo search algorithm, and the fuzzy k-nearest neighbor algorithm.

### 2.1. HSI Color Space

The HSI color space [27] is a very important color space that separates color information from intensity information. The hue (*H*) component represents color information, and the human vision system can distinguish different hues as different colors. The saturation (*S*) component describes color information in terms of how much the hue is diluted with white light—or, in other words, the purity of the color. The intensity (*I*) component represents the amount of light, or the brightness, in an image. One interesting feature of the HSI color system is that it represents colors similar to how they are perceived by humans. The nonlinear transformation from the RGB color space to the HSI color space (*H*, *S*, *I*) is mathematically defined as follows [27]:(1)$$H=\arctan\left(\frac{\sqrt{3}(G-B)}{\left(R-G)+(R-B\right)}\right),\;S=1-\frac{3\min(R,G,B)}{R+G+B},\;I=\frac{R+G+B}{3}$$
where *R*, *G*, *B* ∈ [0, 1] represent the red, green, and blue components of the RGB color space, respectively, while *H* ∈ [0, 2*π*] and *S*, *I* ∈ [0, 1] represent the components of the HSI color space. In this paper, the color components of one pixel in both the RGB and HSI color spaces are considered as feature vectors.

### 2.2. Unsupervised Possibilistic Fuzzy Clustering (UPFC)

UPFC [28,29] is a hybrid fuzzy clustering method that integrates the benefits of fuzzy c-means clustering (FCM) [30] and the possibilistic clustering algorithm (PCA) [31]. FCM is a well-known fuzzy algorithm proposed by Bezdek that works by assigning a fuzzy membership degree to each sample in the dataset, according to a sample’s distances to the cluster centers. A sample will have a high degree of membership to a cluster if it is close to that cluster’s center, or vice versa. One major drawback of FCM is that it is sensitive to outliers or noises in the dataset. Several approaches have been proposed to address the noise sensitivity issue. For example, possibilistic c-means (PCM) [32]—a possibilistic approach to clustering proposed by Krishnapuram and Keller—tried relaxing the probabilistic constraints of FCM to lessen the problem.

Although PCM can improve robustness, some parameter computations still require the implementation of FCM. Yang and Wu [31] also argued that the parameters of PCM were hard to handle and, consequently, proposed another possibilistic clustering approach called the possibilistic clustering algorithm (PCA). PCA is considered to be an improvement over FCM and PCM. While FCM produces memberships, PCA produces possibilities. Its resulting membership function in the form of an exponential function makes it robust to noise; however, it is unfortunately sensitive to the initialization and possible generation of coincident clusters. As a result, UPFC [28] was proposed as an extension of the PCA. It simultaneously generates both memberships and possibilities, and it also overcomes the noise sensitivity problem in FCM and the coincident cluster problem in PCA.

The UPFC algorithm can be briefly described as follows: Let X={x1,x2,…,xNS} be a dataset of NS samples, where each sample is a *p*-dimensional vector (ℝp). Suppose that the number of clusters is NC, and let C={c1,c2,…,cNC} be the set of cluster centers. The goal of UPFC is to minimize the objective function
(2)JUPFC(μ,c)=∑i=1NS∑j=1NC(a⋅μij,FCMm+b⋅μij,PCAn)‖xi−cj‖2+βn2c∑i=1NS∑j=1NC(μij,PCAnlogμij,PCAn−μij,PCAn)
with the constraints
$$\begin{array}{ll} \sum_{j=1}^{N_{C}}\mu_{ij,FCM}=1, & \mathrm{for}\;i=1,2,\ldots ,N_{S}\\ 0\leq \mu_{ij,FCM}\leq 1, & \mathrm{for}\;i=1,2,\ldots ,N_{S}\;\mathrm{and}\;j=1,2,\ldots ,N_{C} \end{array}$$
where $\mu_{ij,FCM}$ and $\mu_{ij,PCA}$ are the fuzzy membership value and the possibilistic value of vector $x_{i}$ in cluster *j*, respectively. The weighting exponents *m* > 1 and *n* > 1 are called the fuzzifier and the typicality, respectively. The multipliers *a* > 0 and *b* > 0 define the relative importance of the fuzzy membership and possibilistic values, respectively. The first term in (2) minimizes the distances from the feature vectors to the cluster centers, while the second term is added for the purposes of partition entropy (PE) and partition coefficient (PC) validity indices [31]. When *b* = 0, UPFC simply becomes PCA.

The objective function (2) can be iteratively solved by Algorithm 1, beginning with the prototype initializations. For each iteration, the fuzzy membership values, the possibilistic degrees, and the cluster centroids are updated using (3), (4) and (5), respectively, until the computed cluster centers converge.
(3)$$\mu_{ij,FCM}={\left[\sum_{k=1}^{N_{C}}{\left(\frac{\left\|x_{i}-c_{j}\right\|}{\left\|x_{i}-c_{k}\right\|}\right)}^{\frac{2}{m-1}}\right]}^{-1},\;\;\;\forall i,j$$
(4)$$\mu_{ij,PCA}=\exp\left(-\frac{bn\sqrt{c}\|x_{i}-c_{j}{\|}^{2}}{\beta }\right),\;\;\;\forall i,j$$
(5)$$c_{j}=\frac{\sum_{i=1}^{N_{S}}\left(a\cdot \mu_{ij,FCM}^{m}+b\cdot \mu_{ij,PCA}^{n}\right)\;x_{i}}{\sum_{i=1}^{N_{S}}\left(a\cdot \mu_{ij,FCM}^{m}+b\cdot \mu_{ij,PCA}^{n}\right)},\;\;\;\forall j$$

Furthermore, as suggested in [31], the parameter β>0 in (2) results in the update Equation (4), where it measures the degree of separation of the dataset. Therefore, it is reasonable to define β as a sample variance computed from the distances between data samples and the vector of sample means ($x_{avg}$) of the dataset, as shown in (6).
(6)$$\beta =\frac{1}{N_{S}}\sum_{i=1}^{N_{S}}\|x_{i}-x_{avg}{\|}^{2},\;\;\;x_{avg}=\frac{1}{N_{S}}\sum_{i=1}^{N_{S}}x_{i}$$

**Algorithm 1.** Unsupervised possibilistic fuzzy clustering (UPFC) [28].
1:Set *c*, *m*, *n*, *a*, *b* where $1<c<N_{C}$, and *m*, *n* > 1, and *a*, *b* > 02:Initiate prototypes, i.e., $c_{j}$, $\mu_{ij,FCM}$, and $\mu_{ij,PCA}$, ∀*i*, *j*3:Compute β using (6)4:
**repeat**
5:Update $\mu_{ij,FCM}$ and $\mu_{ij,PCA}$ using (3) and (4)6:Update the cluster centers using (5)7:**until** prototypes stabilized


### 2.3. Cuckoo Search (CS)

Cuckoo search (CS) [33,34], a nature-inspired metaheuristic algorithm proposed by Yang and Deb in 2009, is based on the obligate brood parasitic behavior of some cuckoo species in combination with the Lévy flights behavior of some birds and fruit flies. Naturally, some cuckoo species deploy an aggressive reproduction strategy by laying their eggs in the nests of other host birds. A host bird may be of another species. If it discovers that the eggs in its nest are not its own, it might throw the alien eggs away, or simply abandon its nest and build a new one. The CS algorithm, along with its variants, provides promising efficiency and has been shown to solve many real-world optimization problems [35,36].

According to [33], three idealized rules of CS can be described as follows:Each cuckoo lays one egg at a time and dumps its egg in a randomly chosen nest.The best nests with high quality of eggs will carry over to the next generations.The number of available host nests is fixed, and the egg laid by a cuckoo is discovered by the host bird with a probability $p_{a}\in [0,1]$. In this case, the host bird can either throw the egg away or abandon the nest and build a completely new nest.

Based on these rules, the generalized pseudocode of CS is summarized in Algorithm 2. Given an objective function to be minimized, each egg represents a solution stored in a nest, and a cuckoo egg represents a new solution. An egg is considered to be of higher quality (i.e., its solution is closer to an optimal value) if it has more resemblance to the host bird’s eggs; thus, it has a higher chance to survive or become the next generation. A Lévy flight is used to model the search for a suitable nest by a cuckoo. The newly laid cuckoo egg (or the new solution) will replace the existing egg in the nest if it has higher quality. Furthermore, a fraction ($p_{a}$) of the worst nests are discarded, and new ones are randomly built.
**Algorithm 2.** Cuckoo search via Lévy flights [33].1:Initialize *N* host nests, $N=\{x_{1},x_{2},\ldots ,x_{N}\}$2:**while** ($t<T_{max}$) **or** (stop criterion) **do**3:  Get a cuckoo randomly by Lévy flights, evaluate its fitness $F_{i}$4:  Randomly choose a nest (say *j*) among *N* nests5:  **if** $F_{j}<F_{i}$
**then**6:     Replace *j* with the new solution7:  **end if**8:  A fraction $p_{a}$ of the worst nests are abandoned, and new ones are built9:  Keep the best solutions10:  Rank the solutions and find the current best11:**end while**

The CS algorithm uses Lévy flights—a type of random walk—to randomly update each cuckoo, with the following update equation:(7)$$x_{i}^{t+1}=x_{i}^{t}+s$$
where $x_{i}^{t+1}$ is the *i*-th cuckoo at generation *t* + 1, and *s* is the step size of the Lévy flights. It can be computed based on Mantegna’s algorithm [37] as follows:(8)$$s=\alpha (x_{i}^{t}-x_{best}^{t})⊕Lé\mathrm{vy}(\beta )$$
(9)$$Lé\mathrm{vy}(\beta )=\frac{u}{|v{|}^{1/\beta }}$$
where α is a constant and β is the Lévy exponent, where 0<β≤2. The operator ⊕ is the element-wise product. $x_{i}^{t}$ and $x_{best}^{t}$ are the *i*-th cuckoo and the best cuckoo at generation *t*, respectively. Lévy(β) is the Lévy distribution, where its parameters u and v are the zero-centered normally distributed random variables, $u\sim N(0,\sigma_{u}^{2})$ and $v\sim N(0,\sigma_{v}^{2})$. Their standard deviations are defined as follows:(10)$$\sigma_{u}={\left[\frac{\Gamma (1+\beta )\cdot \sin(\frac{\pi \beta }{2})}{\Gamma (\frac{1+\beta }{2})\cdot \beta \cdot 2^{(\beta -1)/2}}\right]}^{1/\beta },\;\;\;\sigma_{v}=1$$
where Γ is the standard gamma function.

### 2.4. Fuzzy K-Nearest Neighbor (FKNN)

To assign a feature vector to one prototype, the fuzzy k-nearest neighbor (FKNN) algorithm [38] is intensively applied in this paper. Let a set of vectors C={c11,…,cN11,c12,…,cN22,c1c,…,cNcc} be the multiple prototypes, where cji for j∈{1,2,…,Ni} represents prototype *j* in class *I*, *c* is the number of classes, and Ni is the number of prototypes in class *i*. Given an example x∈ℝp, after the *k*-nearest prototypes of input vector x are found, the membership value of x in class *i*—denoted as ui(x)—can be computed as follows [38]:(11)ui(x)=∑j=1Kuij(1/‖x−xj‖2/(m−1))∑j=1K(1/‖x−xj‖2/(m−1)),   i=1,2,…,Nc
where x1,x2,…,xK are the *k*-nearest prototypes, while *m* is the scaling parameter, whose value is set to 1.5 in this paper. The membership value of prototype xj in class *i* is denoted as uij. Since each prototype belongs to a known class, the membership value uij is simply defined as an indicator function:(12)uij={1,if xj∈class i0,otherwise.
After the membership values of x are obtained for all classes, the classification rule is as follows: x is assigned to class *i* if ui(x)>uj(x) for i≠j.

## 3. Materials and Methods

The proposed automatic system for dental fluorosis classification is based on semantic segmentation of tooth enamel, where each pixel is labeled into five color classes: white, yellow, opaque, brown, and background. The white and light-yellow colors, in general, belong to healthy tooth enamel, while opaque-white and brown colors are considered to be indicators of dental fluorosis in Dean’s index. The background class is assigned to all pixels not in the tooth region. A feature vector was formed by considering a pixel value in both the RGB and HSI color spaces, where the HSI features were computed by (1). The multi-prototype UPFC was used to generate clusters of feature vectors, and the optimal number of clusters was determined by the CS algorithm. One class of colors might consist of multiple clusters. The resulting clusters were used for tooth segmentation to extract a binary mask of the tooth region. Fluorosis severity was graded by evaluating the proportions of opaque and brown pixels in the tooth region. In this paper, four levels of fluorosis are considered, as defined by Yeesarapat et al. [14] in Table 1. The ground truth in the experiments was generated by a D.D.S. dentist with more than 10 years of experience.

### 3.1. Multi-Prototype Generation

Given a set of pixel values from the training images, the multi-prototype UPFC was used to generate clusters of RGB and HSI feature vectors. The overall training process was as presented in Algorithm 3, where the CS algorithm was used to determine the optimal number of clusters ($N_{C}$). The training process started by randomly initializing a set of *N* nests, $N=\{n_{1},n_{2},\ldots ,n_{N}\}$, where $n_{i}$ is the *i*-th nest. Each nest $n_{i}$ contains a randomly chosen integer
(13)$$n_{i}=\mathrm{randint}(\eta ),\;\;\;L_{b}\leq \eta \leq U_{b},\forall i=1,2,\ldots ,N$$
representing the number of clusters between $L_{b}$ and $U_{b}$. The fitness value of a nest $n_{i}$, denoted as $F_{i}$, was defined by the squared error
(14)$$F_{i}={\left(y_{i}-\hat{y}\right)}^{2}$$
where $\hat{y}$ is the expected clustering accuracy and $y_{i}$ is the predicted clustering accuracy for choosing the cluster size $n_{i}$ in the UPFC. Note that each cluster was assigned one label (out of five color labels) according to a majority vote, and the cluster accuracy $y_{i}$ could be computed by comparing the trained clusters with the pixels in the validation set. The experiment described in Section 4.1 used 10-fold cross validation—one fold for training and the remaining folds for validation. Therefore, for *N* nests, we had a set of fitness values, $ℱ=\{F_{1},F_{2},\ldots ,F_{N}\}$. Among all nests, one nest $n_{j}$ was randomly selected to obtain cuckoos with Lévy flights, as well as their fitness $F_{j}$.

A fraction $p_{a}$ of the worst nests were discovered and abandoned according to a binary random vector whose element was $P_{i}\in \{0,1\}$, where
(15)$$P_{i}=\{\begin{array}{ll} 1, & \mathrm{if}\;\mathrm{rand}>p_{a}\\ 0, & \mathrm{otherwise} \end{array}$$
for i=1,2,…,N. Here, rand is a random number in the range [0, 1], and $p_{a}$ is known as the probability of discovery. A nest was rebuilt when $P_{i}=1$; otherwise, it was kept for the next generation. This process was repeated until the minimum fitness of the best nest was less than a specified threshold value ε or the number of iterations exceeded $T_{max}$. Finally, the optimal clusters or multiple prototypes were obtained.
**Algorithm 3.** UPFC via cuckoo search algorithm.1:Initialize *N* host nests, $N=\{n_{1},n_{2},\ldots ,n_{N}\}$2:Randomly choose the number of clusters $n_{i}$ for each nest as shown in (13)3:Get the best nest $n_{i}$ and its fitness $F_{i}$ for the current best nest4:**while** ($t<T_{max}$) **or** ($F_{i}<\varepsilon$) **do**5:   Get cuckoos by Lévy flights using (7) to (10)6:   Calculate the fitness $F_{j}$ using (14) by performing UPFC on each nest. Keep the centroids7:   A fraction $p_{a}$ of the worst nests are abandoned, and new ones are built8:   Calculate the fitness $F_{j}$ using (14) by performing UPFC on each nest. Keep the centroids9:   **if**
$F_{j}<F_{i}$
**then**10:      Replace $F_{i}$ with $F_{j}$ and keep $n_{j}$ as the new best nest11:   **end if**12:**end while**13:**return** the centroid of each prototype

### 3.2. Dental Fluorosis Classification

The multiple prototypes obtained as described in Section 3.1 were used for tooth segmentation to separate teeth from gums. The severity of fluorosis was later classified based on the proportions of opaque and brown pixels in the tooth region.

To perform tooth segmentation, the pixel values of a given image were first computed against all prototypes and then assigned to one of five classes (white, yellow, opaque, brown, and background) using FKNN with *K* = 1 according to (11). Since the tooth region was composed of white, yellow, opaque, and brown pixels, a binary mask of the tooth region could be created. Morphological operators, including opening and dilation, were further used to remove undesired artifacts and enhance the tooth areas. After that, in the resulting binary image, if there were pixels in the tooth area that were misclassified as the background, these pixels would be reclassified again using FKNN with *K* = 5. The segmented binary mask (tooth pixel vs. background) was evaluated for its accuracy as described in Section 4.2.

An extracted tooth region could be used to determine the severity of fluorosis based on the classes of pixel values that fell under the region. Among the four classes, white and yellow pixels are natural colors of normal teeth, so they were considered together as the class of white–yellow pixels. On the other hand, opaque and brown pixels were apparent indicators to quantify fluorosis. As a result, further image segmentation was performed only in the tooth region by considering three classes of pixels: white–yellow, opaque, and brown.

Since fluorosis could be graded by the numbers of opaque and brown pixels appearing in the tooth region, and there might still exist a few tiny areas of opaque or brown pixels after the previous image processing steps, we further processed those tiny objects by removing them if their areas were less than 0.55% of the tooth region for the opaque areas, or less than 0.05% of the tooth region for the brown areas. The resulting tooth segmentation was also evaluated as described in Section 4.2 for pixel accuracy in predicting white–yellow, opaque, and brown pixels.

Upon completion of the image segmentation task, the numbers of white–yellow, opaque, and brown pixels in the tooth region could be determined. As a result, we designed a fluorosis classification rule (Algorithm 4) based on these quantities, where $r_{opaque}$ and $r_{brown}$ are the percentages of opaque and brown pixels in the tooth area, respectively. The idea behind this rule closely followed the enamel description in Table 1, i.e., the size of opaque and brown areas increased as fluorosis became more severe. Furthermore, Stage 2 and Stage 3 fluorosis not only involved brown pixels alone, but were also associated with a reasonably large area of opaque pixels. The choices of parameters used in this rule are discussed in Section 3.3. Its experimental results are shown in Section 4.3. Although this rule was motivated by Yeesarapat et al. [14], there was one key aspect of difference, as discussed in Section 4.4.
**Algorithm 4.** Fluorosis classification rule.1:Given $0\leq \theta_{1}<\theta_{2}<\theta_{3}\leq 1$ and 0≤δ≤12:**function** FluorosisClassifier($r_{opaque}$, $r_{brown}$)3:   **if**
$r_{opaque}\leq \theta_{1}$
**then return** Normal4:   **else if**
$r_{opaque}\leq \theta_{2}$ and $r_{brown}\leq \delta$
**then return** Normal5:   **else if**
$r_{opaque}\leq \theta_{3}$ and $r_{brown}\leq \delta$
**then return** Stage 16:   **else if**
$r_{brown}\leq \delta$
**then return** Stage 27:   **else return** Stage 38:**end if**9:**end function**

### 3.3. Dataset and Parameter Settings

To evaluate the performance of the proposed fluorosis classification algorithm as well as its tooth segmentation steps, we used the dataset of Yeesarapat et al. [14], which was collected by the Intercountry Centre for Oral Health (ICOH), Ministry of Public Health, from children in the rural areas of Chiang Mai province, Thailand. The images were taken with a RICOH Caplio RX camera without any advance preparation. In the study of Yeesarapat et al., experiments were conducted with a total of 22 images, where 7 images and 15 images were assigned for the training and test sets, respectively. In this paper, we used the same 7-image training set for training the UPFC. Furthermore, an additional 113 images were added to the original 15 test set images, resulting in a blind test set of 128 images. Each collected image was taken from each child. Although the number of teeth in each image was different, according to the dentist, this does not affect the performance of the fluorosis stage detection. This is because the detection rule is set based on the treatment condition (mentioned in the Introduction section), where the ratio of the opaque or brown area to the whole teeth area is considered. In addition, this system can be used as a pre-screening system before the patient is sent to see the dentist.

All images were cropped to the mouth region, to clearly show the teeth of each subject, where the final image sizes spanned from 815 × 517 to 1784 × 825 pixels. In addition, these collected images had varied resolutions to show that the proposed system is size- and resolution-invariant. One expert—a dentist with more than 10 years of experience—was asked to grade the severity of fluorosis in all of the images: Normal (no fluorosis), Stage 1 (questionable/mild), Stage 2 (moderate), and Stage 3 (severe). The same expert also provided five classes of pixel-level labels: white, yellow, opaque, brown, and background, where the background encompasses lips, tongues, and gums. The white class also includes reflected light spots in the images.

The parameter settings for the UPFC and cuckoo search in the experiment are shown in Table 2. However, for the fluorosis classifier, the parameters were manually set as follows: $\theta_{1}=0.05$, $\theta_{2}=0.1,\theta_{3}=0.3,\delta =0.007$. The choice of these parameters was based on practical observations. If there was a very small number of opaque pixels (≤5%), or a slightly larger number of opaque pixels (≤10%) together with a very small number of brown pixels (δ≤0.7%), the image would be graded as Normal. As the number of opaque pixels became higher, fluorosis became more severe, as seen by the values $\theta_{1}<\theta_{2}<\theta_{3}$. A large number of opaque pixels ($\theta_{3}>30\%$) but a small number of brown pixels (δ≤0.7%) was classified as Stage 2, while the most severe level was indicated by a large number of brown pixels (δ>0.7%). These parameter settings were based on the treatment condition according to the experienced dentist mentioned in the Introduction section.

The performance [39,40] of the system in pixel-wise and fluorosis classifications was evaluated in term of true positive (TP), true negative (TN), false positive (FP), false negative (FN), true positive rate (TPR), true negative rate (TNR), false positive rate (FPR), false negative rate (FNR), positive predictive value (PPV), negative predictive value (NPV), and accuracy (Acc).

## 4. Experimental Results and Discussion

### 4.1. Experiments on Multi-Prototype Generation

A subset of pixels from the training images, in both the RGB and HSI color spaces, were split into 10 folds. In total, there were approximately 4000, 4000, 2000, 1000, and 1800 selected pixels labeled as white, yellow, opaque, brown, and background, respectively. One fold was used to create multiple prototypes by the UPFC via cuckoo search (Algorithm 3), as described in Section 3.1, while the rest were used for the cluster validation. The parameters involved in these algorithms are described in Table 2. Since each pixel belonged to one of five color classes, each prototype was assigned to a particular class based on a majority vote in that particular cluster. The accuracy of the clustering was evaluated on the remaining nine folds, where we used FKNN with *K* = 1 to predict the label of each pixel. From the experiment, we found that the 10-fold cross-validation accuracy of pixel-wise classification was 91.81%. The performance of the best model from one fold yielded 92.96% pixel-wise accuracy. The number of prototypes generated by this model was 2500, where 662, 610, 414, 257, and 557 prototypes belonged to the white, yellow, opaque, brown, and background classes, respectively. The performances of this model are also presented in Table 3 and Table 4, where we can observe that the value of FNR is particularly high for the opaque class. This is because its colors look relatively similar to the white and yellow classes, posing a challenge for fluorosis classification, since the opaque white color on tooth enamel can be a sign of fluorosis.

### 4.2. Experiments on Tooth Segmentation

The best multi-prototype model obtained from Section 4.1 was used to extract a binary tooth mask for each image, as described in Section 3.2. For each image in the training set, there were 2, 2, 2, and 1 images diagnosed as Normal, Stage 1, Stage 2, and Stage 3, respectively. The segmentation performances of the generated tooth masks for these images are presented in Table 5, where ${\mathrm{Acc}}_{mask}$ is the pixel-wise accuracy of the binary mask, and ${\mathrm{Acc}}_{3}$ is the accuracy of classifying each pixel in the tooth area into three classes: white–yellow, opaque, and brown pixels. Some of the training images and their segmentation results are shown in Figure 1.

For example, image F1_2 in Figure 1b was segmented to 280,326 white–yellow pixels and 22,737 opaque pixels with no brown pixels, resulting in the ${\mathrm{Acc}}_{3}=86.58\%$ correct classification rate in Table 5.

According to Table 5, the ${\mathrm{Acc}}_{mask}$ score for each image is relatively high. It is worth noting that the segmentation accuracy of the training images in Stage 2 (F2_1 and F2_2) is clearly lower than that of the other classes. This is because a portion of the tooth was shaded, causing its appearance to resemble the background class, as depicted in Figure 1c, where the leftmost incisors are segmented as background. The low ${\mathrm{Acc}}_{3}$ score of image F2_2 (53.26%) also exhibited a case of segmentation errors, where a shadow in the image caused a pixel to be classified as a brown pixel.

To further evaluate the segmentation performance of the multi-prototype model, we tested the model with 128 blind test images. The results are shown in Table 5, where ${\mathrm{Acc}}_{mask}$ and ${\mathrm{Acc}}_{3}$ are reported per fluorosis class. Among the 128 images, there were 44, 41, 23, and 20 images diagnosed as Normal, Stage 1, Stage 2, and Stage 3, respectively, as shown in Table 6. The segmentation results on these images demonstrated a good generalization performance of the multi-prototype UPFC model in conjunction with the cuckoo search algorithm, as the average segmentation accuracy was 92.24% and 79.46% for ${\mathrm{Acc}}_{mask}$ and ${\mathrm{Acc}}_{3}$, respectively. The results were also consistent with the results from the training set, reflecting that Stage 2 fluorosis was a challenging case. The reasons for some misclassifications might have been due to reflected light or malocclusion in the image caused by the image acquisition process, or the presence of some artifacts (such as saliva and dental plaque) in the tooth area.

### 4.3. Experiments on Dental Fluorosis Classification

The severity of fluorosis in each image was classified using Algorithm 4, where the percentages of opaque and brown pixels in the tooth area ($r_{opaque}$ and $r_{brown}$, respectively) were obtained from the image segmentation. For example, the segmentation result of image F3_1 (Figure 1d) had 200,431 white–yellow pixels, 117,650 opaque pixels, and 5397 brown pixels; thus, we had $r_{brown}=1.67\%$, as shown in Table 5.

The classification results from the training set and the blind test set are reported in Table 5 and Table 7, respectively. The predicted accuracy on the training set was five images from a total of seven images (71.43%) for the training set. For the blind test set, the performance on each individual class in terms of accuracy was between 81% and 86%, as shown in Table 7. The classification accuracy for all four fluorosis classes was 86 images out of 128 images (67.19%) according to the confusion matrix in Table 8. Some examples of correct prediction of fluorosis classes are provided in Figure 2, where it can be observed that the percentages of opaque pixels (green color) increase from Figure 2a to Figure 2c, with no brown pixels present. The only figure that has brown pixels (red color) is Figure 2d.

In addition, we also investigated misclassifications on the blind test set, some of which are summarized in Figure 3. According to Figure 3a, the misclassification was due to the malocclusion of the right lower incisor, where the shaded area was assigned to brown pixels. In Figure 3b, the expert annotated the pixels in the lower left molar as the white–yellow class; however, our system largely assigned these pixels to the opaque class. This segmentation error caused our system to incorrectly report higher severity of fluorosis as Stage 2, instead of Stage 1.

### 4.4. Comparison with Prior Works

The fluorosis classification rule used by Yeesarapat et al. [14] was rather similar to Algorithm 4. However, one major difference is in Line 3 of our algorithm, which did not appear in Yeesarapat et al.’s. In fact, our paper used the parameters $\theta_{1}=0.05,\;\theta_{2}=0.1,$ $\theta_{3}=0.3,\;\delta =0.007$, while $\theta_{2}=0.05,\;\theta_{3}=0.4,\;\delta =0.0065$ were used by Yeesarapat et al. In order to investigate the importance of Line 3 in Algorithm 4, we illustrate a sample case in Figure 4, where the input image has no sign of fluorosis. According to the results of image segmentation, opaque and brown areas in this image contributed to 1.63% and 1.35% of the entire tooth region, as shown in Figure 4. Consequently, it was misclassified as Stage 3 by Yeesarapat et al. The cause of this misclassification was because pixels in the shaded area behind the lower lip were incorrectly labeled as brown pixels. In fact, any brown area is considered to be sign of fluorosis if there are also a fair number of opaque pixels in the tooth region. Consequently, incorporating Line 3 into our classification rule helped reflect this condition, allowing our algorithm to correctly classify Figure 4 as Normal.

One contribution of our paper over the work of Yeesarapat et al. is in the creation of binary tooth masks through the UPFC, CS, and FKNN algorithms, instead of using manual selection. In addition, to demonstrate another improvement of our method over that of Yeesarapat et al., we compared the classification performance on the 7 training images and 15 blind test images used by Yeesarapat et al., as reported in Table 9. It should be noted that their training images were the same set used in our experiment. Their blind test images were also a subset of the 128 blind test images discussed in Section 4.2 and Section 4.3. The results of fluorosis classification are shown in Table 9, where our model achieved agreement with the expert’s opinion for 5 out of 7 images (71.43%) and 10 out of 15 images (66.67%) for the training and blind test sets, respectively. However, the model of Yeesarapat et al. only achieved agreement for 3 out of 7 images (42.86%) and 8 out of 15 images (53.33%) for the training and blind test sets, respectively. These results could be interpreted as 28.57% and 13.33% improvements for the training and blind test sets, respectively. Consequently, these results demonstrate that our approach to fluorosis classification is superior to that of Yeesarapat et al.

## 5. Conclusions

In this paper, we proposed an automatic system of dental fluorosis classification using six color features from the RGB and HSI color spaces as feature vectors. Unsupervised possibilistic fuzzy clustering (UPFC) was used to cluster these features, since each feature vector had a label in one of five classes—white, yellow, opaque, brown, or background—based on the color of that pixel and whether the pixel was in the tooth area. We applied the cuckoo search algorithm to optimize the numbers of clusters in each class. A set of pixels from seven training images, trained with these algorithms, resulted in a set of optimal prototypes, where tooth segmentation was then performed using the fuzzy k-nearest neighbor (FKNN) algorithm, together with some morphological operations. We classified the stages of fluorosis into Normal, Stage 1, Stage 2, and Stage 3, based on the proportions of opaque and brown pixels in the tooth area. Our experimental results showed that the proposed method was superior to prior works, as it correctly classified four fluorosis classes for 86 out of 128 images in the blind test set. In addition, the average pixel accuracy of the segmented binary tooth masks was 92.24%, and the average pixel accuracy of teeth segmented into white–yellow, opaque, and brown pixels was 79.46%. 

In future works, it will be possible to improve the performance of the proposed method by replacing the fluorosis classification rule with a learning algorithm. Increasing the size of the training set—especially with more variations in fluorosis conditions and light conditions—would also be another way to enhance the performance. Moreover, to improve the efficiency of the system, the results need to be confirmed with clinical tests. Otherwise, the fluorosis stage detections from the system might be mistaken for other enamel hypoplasias or pigmentations due to drugs or smoking, etc. In addition, to increase the performance of the application from the point of view of a homogeneous application, we could develop a system that requires a captured image to cover the vestibular surface of at least four upper and lower incisors.

As our proposed model is lightweight, one practical direction would be to deploy our model to mobile devices and evaluate its real-world performance in both clinical and non-clinical settings.

## Figures and Tables

**Figure 1 ijerph-20-03394-f001:**
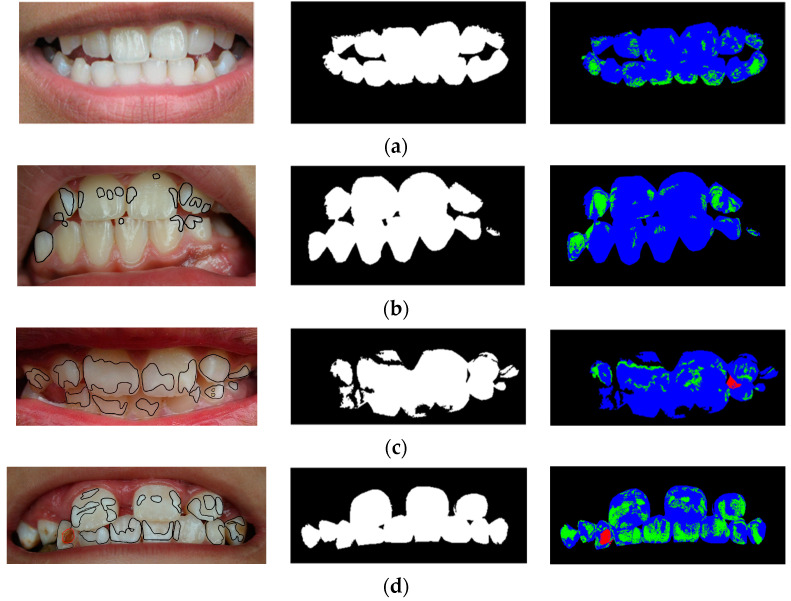
Examples of each fluorosis class from the training set and their segmented images: (**a**) Image N2 (Normal); (**b**) Image F1_2 (Stage 1); (**c**) Image F2_2 (Stage 2); (**d**) Image F3_1 (Stage 3). The left column presents the expert’s labels of opaque pixels and brown pixels, encircled in black and red, respectively. The center column presents the predicted binary tooth masks. The right column presents the predicted white–yellow, opaque, and brown pixels in blue, green, and red colors, respectively.

**Figure 2 ijerph-20-03394-f002:**
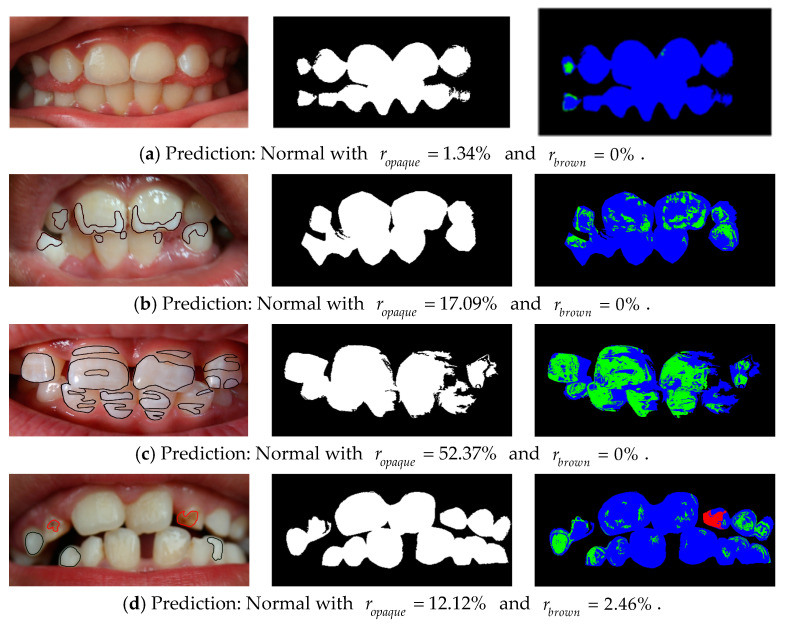
Examples with correct prediction of each fluorosis class from the blind test set and their segmented images: (**a**) Normal; (**b**) Stage 1; (**c**) Stage 2; (**d**) Stage 3. The left column presents the expert’s labels of opaque pixels and brown pixels, encircled in black and red, respectively. The center column presents the predicted binary tooth masks. The right column presents the predicted white–yellow, opaque, and brown pixels in blue, green, and red colors, respectively.

**Figure 3 ijerph-20-03394-f003:**
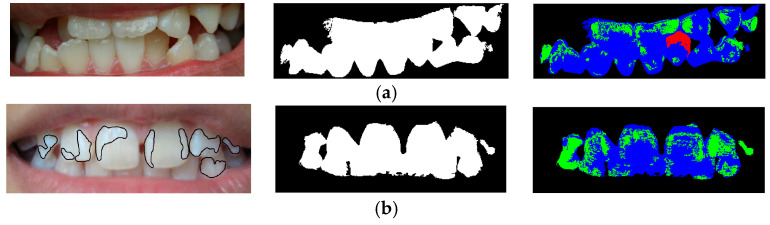
Examples of misclassification on the blind test set and their segmented images. (**a**) Expert’s opinion: Normal; Prediction: Stage 3 with $r_{opaque}=19.84\%$ and $r_{brown}=3.23\%$. (**b**) Expert’s opinion: Stage 1; Prediction: Stage 2 with $r_{opaque}=33.95\%$ and $r_{brown}=0\%$.

**Figure 4 ijerph-20-03394-f004:**
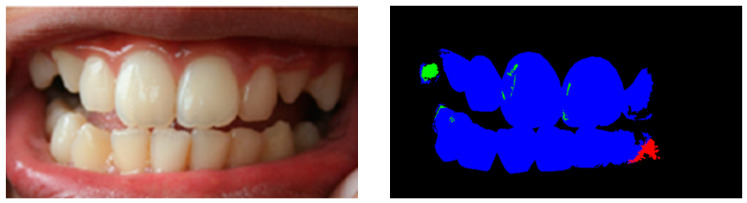
An image in the class Normal was misclassified as Stage 3 if Line 3 in Algorithm 4 was removed, because the segmented image has $r_{opaque}=1.63\%$ (green color) and $r_{brown}=1.35\%$ (brown color).

**Table 1 ijerph-20-03394-t001:** Severity of dental fluorosis according to Dean’s index and Yeesarapat et al. [14].

Dean’s Index	Description of Tooth Enamel	Yeesarapat et al.
Normal	Smooth, glossy, pale creamy-white translucent surface.	Normal
Questionable	A few white flecks or white spots that cannot be determined as very mild or normal.	Stage 1
Very mild	Small, opaque, paper-white areas covering less than 25% of the tooth surface. This also includes teeth showing no more than about 1–2 mm of white opacity at the tips of the summits of the cusps of the bicuspids or second molars.	Stage 1
Mild	Opaque white areas covering less than 50% of the tooth surface.	Stage 1
Moderate	All tooth surfaces affected; marked wear on biting surfaces; brown stains may be present.	Stage 2
Severe	All tooth surfaces affected; discrete or confluent pitting; brown stains present.	Stage 3

**Table 2 ijerph-20-03394-t002:** List of UPFC and cuckoo search parameters.

Algorithm	Parameter	Description	Value
UPFC	*m*	Fuzzifier	1.5
	*n*	Typicality	1.5
	*a*	Relative importance of fuzzy membership	0.5
	*b*	Relative importance of possibilistic value	0.5
Cuckoo Search	*N*	Number of nests	30
	$\left[L_{b},U_{b}\right]$	Lower/upper bounds of the number of clusters	[1000, 2500]
	$\hat{y}$	Expected accuracy of clustering	0.99
	α	Step size scaling factor	0.1
	β	Lévy exponent	1.5
	$p_{a}$	Discovery probability	0.25
	ε	Tolerance	0.001
	$T_{max}$	Maximum number of generations	100

**Table 3 ijerph-20-03394-t003:** Performance of pixel-wise classification based on multiple prototypes.

Class	TP	FN	FP	TN	TPR	TNR	FPR	FNR	PPV	NPV	ACC
White	405	21	54	856	95.07	94.07	5.93	4.93	88.24	97.61	94.39
Yellow	386	18	33	899	95.54	96.46	3.54	4.46	92.12	98.04	96.18
Opaque	181	37	3	1115	83.03	99.73	0.27	16.97	98.37	96.79	97.01
Brown	96	7	4	1229	93.20	99.68	0.32	6.80	96.00	99.43	99.18
Background	174	11	0	1151	94.05	100.00	0.00	5.95	100.00	99.05	99.18

**Table 4 ijerph-20-03394-t004:** Confusion matrix of pixel-wise classification based on multiple prototypes.

Actual Class	Predicted Class				
	White	Yellow	Opaque	Brown	Background
White	405	18	3	0	0
Yellow	18	386	0	0	0
Opaque	31	4	181	2	0
Brown	4	3	0	96	0
Background	1	8	0	2	174

**Table 5 ijerph-20-03394-t005:** Segmentation performance on each image in the training set, the percentages of opaque and brown pixels, and the comparison between predicted fluorosis stage vs. expert’s opinion.

ID	Expert’s Opinion	${\mathrm{Acc}}_{mask}(\%)$	${\mathrm{Acc}}_{3}(\%)$	$r_{opaque}(\%)$	$r_{brown}(\%)$	Prediction
N1	Normal	94.54	92.75	7.25	0.00	Normal
N2	Normal	96.03	84.93	15.07	0.00	Stage 1
F1_1	Stage 1	90.05	82.18	12.59	0.00	Stage 1
F1_2	Stage 1	95.02	86.58	11.16	0.00	Stage 1
F2_1	Stage 2	78.34	62.63	37.85	0.00	Stage 2
F2_2	Stage 2	78.62	53.26	9.20	1.19	Stage 3
F3_1	Stage 3	98.40	69.64	36.37	1.67	Stage 3

**Table 6 ijerph-20-03394-t006:** Segmentation performance on 128 images in the blind test set.

Expert’s Opinion	No. of Images	${\mathrm{Acc}}_{mask}(\%)$	${\mathrm{Acc}}_{3}(\%)$
Normal	44	91.97	91.57
Stage 1	41	93.71	79.55
Stage 2	23	89.52	67.74
Stage 3	20	92.93	66.08
Average	-	92.24	79.46

**Table 7 ijerph-20-03394-t007:** Fluorosis classification accuracy on 128 blind test images.

Class	TP	FN	FP	TN	TPR	TNR	FPR	FNR	PPV	NPV	ACC
Normal	34	10	13	71	77.27	84.52	15.48	22.73	72.34	87.65	82.03
Stage 1	26	15	9	78	63.41	89.66	10.34	36.59	74.29	83.87	81.25
Stage 2	15	8	11	94	65.22	89.52	10.48	34.78	57.69	92.16	85.16
Stage 3	11	9	9	99	55.00	91.67	8.33	45.00	55.00	91.67	85.94

**Table 8 ijerph-20-03394-t008:** Confusion matrix of fluorosis classification on 128 blind test images.

Actual Class	Predicted Class			
	Normal	Stage 1	Stage 2	Stage 3
Normal	34	5	2	3
Stage 1	9	26	2	4
Stage 2	4	2	15	2
Stage 3	0	2	7	11

**Table 9 ijerph-20-03394-t009:** Comparison of the proposed fluorosis classification approach with that of Yeesarapat et al. [14] on 7 training images and 15 blind test images.

ID	Set	Expert	Prior Work	Ours	ID	Set	Expert	Prior Work	Ours
N1	Train	Normal	Stage 1	Normal	F1_7	Test	Stage 1	Stage 1	Stage 1
N2	Train	Normal	Stage 1	Stage 1	F2_1	Train	Stage 2	Stage 1	Stage 2
N3	Test	Normal	Normal	Normal	F2_2	Train	Stage 2	Stage 3	Stage 3
N4	Test	Normal	Normal	Normal	F2_3	Test	Stage 2	Normal	Normal
N5	Test	Normal	Normal	Normal	F2_4	Test	Stage 2	Stage 1	Stage 3
F1_1	Train	Stage 1	Stage 1	Stage 1	F2_5	Test	Stage 2	Stage 2	Stage 2
F1_2	Train	Stage 1	Stage 1	Stage 1	F2_6	Test	Stage 2	Stage 1	Stage 1
F1_3	Test	Stage 1	Stage 3	Stage 3	F2_7	Test	Stage 2	Stage 1	Stage 2
F1_4	Test	Stage 1	Stage 2	Stage 2	F3_1	Train	Stage 3	Stage 3	Stage 3
F1_5	Test	Stage 1	Stage 1	Stage 1	F3_2	Test	Stage 3	Stage 1	Stage 3
F1_6	Test	Stage 1	Stage 1	Stage 1	F3_3	Test	Stage 3	Stage 3	Stage 3

## Data Availability

Due to privacy and ethical issues, the data sets cannot be publicized.

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
