# Peer review of "Fuzzy K-Nearest Neighbor Based Dental Fluorosis Classification Using Multi-Prototype Unsupervised Possibilistic Fuzzy Clustering via Cuckoo Search Algorithm"

_ijerph, 2023, doi:10.3390/ijerph20043394_

Round 1

Reviewer 1 Report

It is a manuscript with many deficiencies from the clinical methodological. There is a low understanding or management of the concept of fluorosis, their analysis is based only on spots, however, in moderate or severe Dean´s stages, comparable to 2 and 3 of Yeesarapat, there are cavitations that represent loss of mineralized tissue, which does not consider in their methodology. His knowledge of the disease should improve, for example, he considers that the problem of fluorosis is mostly aesthetic (page 2/line 50), however, in higher degrees the confluent pitting could be a factor that facilitates pulpal disorders.

The manuscript has consistency and writing errors, for example, in the abstract page 1/line 22, it mentions that they will be grouped into six categories when it only mentions 5 (white, yellow, opaque, brown, and background.). As well as format errors and omissions since there are acronyms that do not explain their meaning (table 3).

The methodology has significant weaknesses or deficiencies in the sample.

1. The homogeneity in the number of teeth, as well as in their extension in the image is not homogeneous, in some photos are observed 6 teeth, in another 4, in some images the complete vestibular face is seen in others only 2 /3 part (fig. 1).

2. Indicate a wide range regarding the size of the images (page 10/line 331), but not the minimum resolution of the images in dpi's.

3. They do not indicate whether these 128 images correspond to 128 patients, the lack of clinical data means that important correlations cannot be established.

4. He mentions that an expert assigned fluorosis degrees to the images, however, to really assign a grading the expert must be calibrated and make the estimation in in a real patient, not images, it is always necessary to observe the pattern of fluorosis in the first molars, since otherwise the diagnosis can be misdiagnosed by confusing fluorosis with enamel hypoplasias or pigmentations due to drugs.

5. An important and central problem is that the algorithm for the segmentation of the tooth and their execution of the generates errors in the classification, which must be corrected. Parameters such as malocclusion, or number or extension of the teeth can influence a bad classification. These characteristics are, in my opinion, what causes ACC3 to be lower in relation to the degree of fluorosis.

Author Response

Please see the attached pdf file.

Reviewer 2 Report

Fuzzy K-Nearest Neighbor Based Dental Fluorosis Classification Using Multi-prototype Unsupervised Possibilistic Fuzzy Clustering via Cuckoo Search Algorithm

This paper describes a novel approach for automated image segmentation, identification and classification of the severity of enamel lesions occurring in dental fluorosis.

The method description, its optimization using a training  data set and its validation on a blind test set appear to have been performed with rigor, and the presented results appear tied to the conclusions in a plausible way. Nevertheless, given the medical background of this reviewer, programming and mathematical soundness of the proposed method should also be evaluated by a reviewer with a background in engineering & programming.

Some revisions are suggested by this reviewer, referring to the highlighted parts of the attached marked manuscript:

Lines 39-40: Consider replacing “Its prevalence occurs” with “Fluorosis occurs”, since the term “prevalence”, in this context, denotes the frequency of fluorosis, which in itself cannot “occur”

Line 40: Consider replacing “the global prevalence over” with “a global prevalence of over”.

Line 45: Consider replacing “the amount of dose” with “the dose”, to avoid a pleonasm.

Line 47:  Consider replacing “risk indicator” with “risk factor”, as it more accurately describes the role played by swallowing fluoridated toothpastes by children.

Line 50: Consider adding “mild” before “dental fluorosis”, since moderate and severe fluorosis may turn tooth enamel brittle, leading to marked tooth-wear, as subsequently specified in Table 1.

Lines 57-58: Consider adding “by” after “developed”.

Table 1: Consider adding spaces or horizontal lines between index categories, to help readers distinguish between the corresponding descriptions of tooth enamel.

Line 98: Please correct the misspelled word “fluorois”.

Lines 124, 126, 354, 363, 364, 365, 366, 368, 375, 377, 385-398, 408, 410, 411-435, 441, 446, 447-465, 483-497: Please use past tense (not present tense) when presenting the aim, the employed methods or the obtained results of the current study. Past tense improves writing precision, given that such statements refer to what has been planned/performed/found in a concluded study, as opposed to statements that use present tense, which should only be used to communicate generally accepted knowledge or generalizable conclusions supported by strong evidence.

Line 131: Results should preferably be separated from discussions (as distinct sections of a scientific article). However, this correction may be optional if the journal editors accept their combined presentation in the case of this article, given the technical nature of many results, which may require immediate discussion and supplementary explanations.

Line 247: Consider removing “are”, since it duplicates “belong”.

Figures 1, 2 and 3: Would it be possible to highlight the fluorosis stains identified by the expert by only delimiting their perimeters with dotted lines (not by covering them completely), so that readers can see the actual fluorosis lesions? Also, please specify who the diagnosing expert was and his/her qualification(s) for performing this medical act. Please do that in section Material and Methods, under a similar context to the one described in line 331, but transferred to section Material and Methods, since information regarding the experimental method should not appear in section Results.

Please verify and transfer to Material and Methods all other paragraphs that describe how the experiment was conducted instead of presenting strictly results.

Line 359: Consider adding “in” before “these”.

Line 523: Please correct the repetition.

Author Response

Please see the attached pdf file.

Round 2

Reviewer 1 Report

The inclusion of a DDS within the analysis directed the manuscript in a better way, helped to recognize the limitations of the study, a situation that they recognize page 13 line 456-458 “some misclassifications might be due to reflected light or malocclusion in the image caused by an image acquiring process or there were some artifacts such as saliva and dental plaque in the tooth area”. It is true that to establish the grading a small area on the surface of the tooth is enough, however, as improvement measures and establish a homogeneous application of the algorithm, it could be necessary to consider as a basic criterion that the photographs present the entire vestibular surface at least the 4 upper and lower incisors, this point could be included in the discussion added to its conclusion page 17 line 578-581.

Author Response

Please see attached pdf file.
